# Exosome-Based Vaccines: Pros and Cons in the World of Animal Health

**DOI:** 10.3390/v13081499

**Published:** 2021-07-29

**Authors:** Sergio Montaner-Tarbes, Lorenzo Fraile, María Montoya, Hernando Del Portillo

**Affiliations:** 1Innovex Therapeutics SL, 08916 Badalona, Spain; hadelportillo@innovexther.com; 2Agrotecnio Center, Department of Animal Science, University of Lleida, 25198 Lleida, Spain; lorenzo.fraile@ca.udl.cat; 3Centro de Investigaciones Biológicas Margarita Salas (CIB-CSIC), Ramiro de Maeztu 9, 29040 Madrid, Spain; maria.montoya@cib.csic.es; 4ICREA at ISGlobal, Barcelona Institute for Global Health and IGTP, German Trias I Pujol Health Research Institute, 08916 Badalona, Spain

**Keywords:** extracellular vesicles, vaccines, viral diseases

## Abstract

Due to the emergence of antibiotic resistance and new and more complex diseases that affect livestock animal health and food security, the control of epidemics has become a top priority worldwide. Vaccination represents the most important and cost-effective measure to control infectious diseases in animal health, but it represents only 23% of the total global animal health market, highlighting the need to develop new vaccines. A recent strategy in animal health vaccination is the use of extracellular vesicles (EVs), lipid bilayer nanovesicles produced by almost all living cells, including both prokaryotes and eukaryotes. EVs have been evaluated as a prominent source of viral antigens to elicit specific immune responses and to develop new vaccination platforms as viruses and EVs share biogenesis pathways. Preliminary trials with lymphocytic choriomeningitis virus infection (LCMV), porcine reproductive and respiratory syndrome virus (PRRSV), and Marek’s disease virus (MDV) have demonstrated that EVs have a role in the activation of cellular and antibody immune responses. Moreover, in parasitic diseases such as *Eimeria* (chickens) and *Plasmodium yoelii* (mice) protection has been achieved. Research into EVs is therefore opening an opportunity for new strategies to overcome old problems affecting food security, animal health, and emerging diseases. Here, we review different conventional approaches for vaccine design and compare them with examples of EV-based vaccines that have already been tested in relation to animal health.

## 1. Introduction

According to the Food and Agriculture Organization (FAO), one of the key challenges of the future will be covering food demand [1]. The United Nations estimates that the world population will grow to 9.7 billion by the year 2050 and to 10.8 billion by 2080, representing an increase of 32% and 47%, respectively. In addition, reports from the WHO/FAO have indicated that meat and milk consumption will increase by more than 44% by 2030 [2]. Hence, there are major challenges to face in the following years, such as improving agricultural productivity and preventing transboundary, emerging agriculture, and food system threats [3]. Unfortunately, there are few studies regarding the economic impact of the transboundary diseases of livestock, and most of them are only related to production costs. Other features, such as price and market effects, trade, food security, nutrition, the financial costs of outbreaks, and monitoring and control measures, have not been properly evaluated [4,5]. Due to globalization and the wide economic and social impacts, the control of diseases for food security and animal health has become a top priority of the WHO/FAO [3,4]. Some of these transboundary livestock animal diseases are brucellosis, bovine tuberculosis, bovine spongiform encephalopathy (BSE), influenza, foot and mouth disease (FMD), peste des petits ruminants, classical or African swine fever (CSF/ASF), and porcine reproductive and respiratory syndrome (PRRS).

Vaccines are one of the most cost-effective tools to control and eventually eliminate infectious diseases, and they are a basic strategy of preventive medicine programs in livestock. In this context, the global animal health product market was worth USD 15 billion in 2005 [6] and of this amount, vaccines represented only 23% of the total [7], thus indicating that a concerted effort on the research on and the development of new vaccination strategies for novel and old veterinary infectious diseases is desperately needed. There are indeed a wide range of pathogens for which no vaccines are presently available [8], including viral diseases, and according to the OIE, animal health comprises more than half of them. They pertain to 22 families of viruses in which herpesvirus, rhabdovirus, poxvirus, and paramyxovirus are responsible for a high percentage of registered infections [9]. Here, we review different conventional approaches for vaccine design and compare them to the few examples of extracellular vesicles (EVs)-based vaccines that have already been tested in relation to animal health.

## 2. Live Attenuated Vaccines

Since the rinderpest vaccination campaign, classical vaccinology methods have been applied to control viral infections, focusing on the use of killed and attenuated vaccines [8]. The first example of an effective attenuated vaccine in veterinary medicine was developed by Walter Plowright in the 1960s [10]. This vaccine was based on the rinderpest virus that had been repeatedly passaged in calf kidney cells in vitro. After 40 passages, the virus lost part of their virulence, and when used to immunize cattle, no significant increase in temperature or adverse reaction related to inoculation was observed; most importantly, this vaccination approach conferred immune protection against a challenge with a virulent strain [10]. In addition, serum neutralizing titers were even detected at 36 months post-vaccination in contrast to 90 passages, in which the protective immune capacity was partially lost [8]. Attenuated viral strains, however, could revert to their initial virulence after few passages in cattle, constituting one of the main drawbacks of this vaccination approach. Partial protective immunity upon vaccination has also been observed in other attenuated viral veterinary vaccines. For example, in the case of PRRS virus (PRRSV), there are several approved and commercially available live attenuated virus (MLV) vaccines [11], but they have unpredictable efficacy depending on the vaccination group (fattening pigs, sows, or gilts) and could not confer effective cross-protection among antigenically and genetically different strains without the risk of reversion to a virulent state or recombination with wild-type strains [11,12]. An important key gap in vaccination for PRRSV is that neither the crucial antigens nor the mechanisms of how protective immune responses are triggered have been identified [13]. The main concern of biosafety also applies to other diseases, such as ASF, but in this case, other approaches, such as genome editing, have been applied to delete virulent factors from ASF viruses, resulting in a defective replication or attenuated strains with increased safety (Table 1) [14]. Not all viral diseases can be safely controlled using live attenuated vaccines, and alternative strategies are required.

## 3. Inactivated and Subunit Vaccines and Adjuvants

Inactivated vaccines consist of a pathogen that is usually frown in cell culture, and after, it is inactivated by physical or chemical methods that cause the denaturation of proteins or nucleic acids, keeping most of the original structure but losing the infectiveness and replication capacity [8]. Many examples of inactivated vaccines are found in the literature in which the activation of immune responses (cellular and antibody-mediated) are characterized, and different degrees of protection are achieved [31,32], including examples such as inactivated vaccine candidates against capripox virus (CaPV), which have sterile immunity when combined with specific adjuvants [33]. The main advantage of inactivated vaccines is that they offer a good safety profile with short to mid-term protection. Nevertheless, for viral diseases, there is no long-term protection because the replication of the pathogen is destroyed by the inactivation method. In addition, as many inactivated vaccines are unable to generate protection against newly emerged field strains, those vaccines need to be updated to cover new strains each time [8].

Subunit vaccines typically use different expression platforms, such as bacteria, yeast, insect, and mammalian cell cultures, to produce the antigen of interest. More recently, other strategies based on non-fermentative systems (plants or insects) have also been used [9]. Classically, subunit vaccines are classified as inactivated vaccines in which a part of a pathogen is used to generate an immune response. They exhibit an additional advantage of being safer in comparison to live attenuated vaccines, and also allow higher production rates (scalability) than those of whole-pathogen vaccines [17,34]. However, the main limitations, such as weak immune responses and the need for several doses (boosts), make this kind of vaccine strategy more expensive and less competitive than others.

Other strategies combine several epitopes to increase the antigenicity and effectiveness of subunit vaccines. A good example is that based on the construction of dendrimeric peptides, generally termed B_n_T, in which several copies of B-cell epitopes are combined with T-cell epitopes via a connection through a lysine core [35]. This approach has been tested with epitopes from the CSF and FMD viruses (CSFV and FMDV), with great success in eliciting Th1 responses, at least in the case of FMDV (high number of IFN-γ producing CD4^+^ T cells that also produced TNFα), and a high anti-peptide antibody response with some constructs showing neutralizing antibody activities [35,36,37].

Due to the limitations of subunit and inactivated vaccines, adjuvants are normally added to stabilize the antigen and/or to induce the activation of the innate immune cell signaling pathways with the aim of generating stronger, long-lasting, and faster immune responses to the antigen of interest (Table 1) [38,39]. As a consequence, depending on the nature of the pathogen, such as bacteria, parasites or viruses, selection of the correct adjuvant to enhance immune responses is a key aspect to elicit protective immune responses. Adjuvants could act as carriers of subunit proteins, as inductors by means of inducing danger signals (pathogen-associated molecular patterns or PAMPs), or as potentiators of immune responses, such as Toll-like receptor molecules [40]. Depending on the target animal species e.g., poultry [41], swine [42], or ruminants [43], different kinds of molecules have been tested as adjuvants. They include oil in water emulsions (Montanide^®^), aluminum, and other mineral salts [44]. This field is therefore one of the most promising and yet less explored fields regarding the generation of effective vaccine formulations.

## 4. Recombinant DNA and Viral-Vectored Vaccines

Nowadays, DNA technology and viral vectors have changed the way in which immunity can be achieved. Direct antigen production in the target species with the same post-translational modifications, thus mimicking properties and features of a real infection without causing pathology, is a major advantage.

DNA vaccines are used as carriers for the coding gene of interest. Cells around the injection site uptake the plasmid DNA, activating both cellular and antibody immune responses. In addition, methods for DNA technology are well established and allow the modification of the inserted genes to encode the proteins from new viral strains or re-emerging diseases [45]. The main advantages of this kind of vaccination are the easy production and design, allowing, in some cases, the differentiation of infected animals from those vaccinated (DIVA), one of the most important features for transboundary livestock disease vaccines. Moreover, combining different antigens to enhance the immune response (as well as the humoral and cellular responses) and to create multivalent vaccines against different pathogens is an attractive possibility. The main disadvantage of this approach in livestock is the quantity of DNA necessary to induce a good immune response due to the large size of farm animals (swine, cattle, goats, among others). This situation could be improved by using adjuvants or stronger promoters for gene expression [9]. Some examples include heterologous combinations following a prime–boost strategy using, for example, DNA and MLV for PRRSV (enhancing T cell and antibody responses) [46]. In the case of the ASF virus (ASFV), a combination of multiple proteins [47] and DNA antigens have been used, as explained in detail elsewhere [14]. Other viruses, such as the West Nile virus (WNV), FMDV, the infectious bursal disease virus, and the Aujeszky’s disease virus, follow the same trend [7,17].

Viral-vectored technology for veterinary vaccines is based on models widely used for human diseases and include adenovirus, herpesvirus, and poxvirus vectors. As an example, the canarypox virus system has been used as a platform for a wide range of veterinary vaccines, including WNV, the canine distemper virus, the feline leukemia virus, the rabies virus, and the equine influenza virus [8]. This strategy has several advantages, including the construction of multivalent vaccines, a generation of long-lasting immunity, low cost, ease of production, immunogenicity even when administered by different immunization routes, high humoral and cellular immune responses, intrinsic adjuvant properties, and the induction of mucosal immunity (Table 1) [48].

Within this category of recombinant viral vectors, there are two main types: replicating viral vectors, in which viruses have all of the necessary components to replicate inside the host cells, and non-replicating viral vectors, which are deficient in some of the viral functions essential for the replication and/or the assembly of new viral particles [45]. Some licensed vaccines used in veterinary medicine include the canarypox viral vector for different diseases, such as equine influenza, the canine distemper virus (haemagglutinin), the WNV envelope protein and rabies virus glycoprotein G, the Newcastle disease virus (LaSota strain) viral vector expressing avian influenza H5 for prevention of Newcastle virus disease and avian influenza (Avimex^®^), and the vaccinia virus expressing the rabies virus glycoprotein G (Boehringer Ingelheim Animal Health). However, the risk of developing immunity against each viral vector due to repeated immunizations, reversion to virulence, or the release of genetic modified organisms (GMOs) in the community need to be considered.

## 5. Nanovaccines: The New Era of Vaccination

Nanoparticulate vaccine formulations have emerged as new antigen delivery system that could sustain gradual release of immunogens, activating cell-mediated and antibody responses both locally in the site of injection and systemically [31]. Different strategies are included in this category, such as virus-like particles (VLPs) and nanocarriers made of polymers, liposomes, and more recently, EVs (outer membrane vesicles, microvesicles and exosomes), in which safety and good immunogenic properties are combined to obtain better vaccines. When compared to classical subunit vaccines (easy to produce in large quantities but mostly needing adjuvants to improve immunogenicity), nanovaccines offer better immune responses because particulate antigens, such as virosomes, VLPs, polymeric nanocarriers, liposomes, and among other things, exhibit large surfaces with improved receptor-interacting properties for antigen-presenting cells when compared to soluble proteins [49,50].

VLPs are particles in the size range of 25–100 nm in diameter with a similar structure to that of a virion but without the capacity of self-replication in target cells, bearing only antigens of interest that are either exposed on the surface or inside the viral particle. VLP technology is based on the natural features of viral components, such as the major proteins in the capsid and envelope components, to self-assemble spontaneously forming virions when expressed in different cellular systems [51]. Empty virions that are free of genetic material offer a safer strategy for vaccinations, as most of the time, they do not require adjuvants due to their nanosize because they easily reach the lymph nodes to interact with B-cells and the dendritic cells [50,52]. Moreover, VLPs are scalable under good manufacturing practice (GMP) conditions using varying strategies, including yeast, mammalian, and insect cell systems (Table 1). Several studies have been conducted regarding the development of VLPs against different animal diseases with different outcomes, including PRRSV [53,54], swine influenza A virus [55], Newcastle virus [56], different fish viral diseases [57], and FMDV in cattle [58] and pigs [59].

## 6. Extracellular Vesicles (EVs)

EVs represent a new and important source of antigens and molecules related to immune responses that can be exploited for human and, more recently, for animal vaccinations [29]. EVs are lipid bilayer nanovesicles produced by almost all living cells, including prokaryotes and eukaryotic cells. They were first discovered in studies addressing the fate of the transferrin receptor during the maturation process of red blood cells [60,61]. Later, it was shown that EVs released by B-cells contained MHC-class II molecules on their surface and were capable of inducing antigen-specific MHC class II-restricted T cell responses [62]. This seminal work demonstrated that EVs could have an important role in antigen presentation in vivo and paved the way for their use in novel therapeutic approaches.

EVs vary in size from 20 to 1000 nm in diameter depending on different factors, such as producer cells and biogenesis mechanisms [63]. Moreover, they can be classified according to the biogenesis process into three main categories as (i) apoptotic bodies (produced by cell death), (ii) microvesicles if the direct origin is the budding of the plasma membrane, and (iii) exosomes if they originated inside the cell in multivesicular bodies (MVB, endocytic compartment), and they are then secreted to the extracellular space by the fusion of the MVB with the plasma membrane. There are several markers associated with EVs, such as tetraspanins (CD9, CD63, CD81 and others); heat shock proteins (HSP70 and HSP90) and some 14-3-3 proteins; major histocompatibility complex molecules (MHC-I and MHC-II); and enzymes, including GAPDH and enolase-1 [29,64].

One of the main features of EVs is that protein and nucleic acid cargo reflects the cell of origin, generating a heterogeneous population when biofluids (serum, plasma, urine, among others) are analyzed. By reflecting the cell of origin, EV content can be related to normal and pathological conditions [65]. For example, Epstein–Barr virus infected cells release EVs containing the LMP-1 protein as a mechanism for immune evasion and virus survival [66]. Another example is the hepatitis C virus (HCV), which uses CD81^+^ EVs to cloak the viral RNA to be released into the extracellular space, thus avoiding detection by immune cells and neutralization mechanisms [67]. In addition, viral entry receptors can be transferred to non-susceptible cells (such as CCR5 receptor in HIV) by means of EVs allowing later infection by HIV-1 [68]. Furthermore, Ebola virus VP40^+^ exosomes induce cell death and apoptosis. The Ebola virus VP40 protein itself can trigger a change in the proteomic cargo of extracellular vesicles (including cytokines), which influences pathogenesis and causes decimation of immune cells [69]. In conclusion, as viruses are able to take advantage of the EV biogenesis pathway to modify their protein cargo and manipulate immune responses, it is important to evaluate EVs as a prominent source of viral antigens that could be exploited for vaccination. There is an increase in research taking advantage of this feature based on characterizing the proteins and nucleic acids associated with EVs from different pathologies in order to discover biomarkers and novel antigens (Figure 1). However, the development of better methods to scale-up production, quantification, characterization, and separation methods remains in progress. In the near future, these will allow the capture of EVs containing a particular pathogen signature to activate the desired immune response in the host or to detect a particular biomarker for a given disease [70].

The first report of using of EVs in a vaccination trial for animal diseases was conducted in chickens exposed to *Eimeria tenella*. Here, CD45^+^ dendritic cells were isolated from the intestine and pulsed ex vivo with *Eimeria* antigens from the sporozoites. Isolated and characterized EVs showed that proteins such as MHC-I and MHC-II, CD80, flotillin and HSP70, were present at their surface. Moreover, after injection with EVs, the animals exhibited a higher number of cells (from the cecal tonsil and spleen) expressing IgG or IgA antibodies against *E. tenella* antigens [71]. In addition, a higher number of IL-2, IL-16, and IFN-γ producing cells were elicited when compared to those animals vaccinated with the antigen alone. After the challenge, they exhibited reduced oocyst shedding, less intestinal lesions, lower mortality, and increased body weight gains.

Research using EVs in veterinary viral diseases is not abundant, and vaccination trials are less common when compared to those using other pathogens, such as parasites and bacteria. This effect is due to the fact that viral replication inside the cell shares EV biogenesis pathways; thus, confounding results could be obtained, as EVs and viruses have similar sizes and densities that make separation difficult when both are present in the host during acute infection (Figure 1) [72,73,74]. However, some examples can be found in the literature where this situation has been addressed, and these are presented below.

The first vaccination trial using animal virus and EVs used dendritic cell-derived exosomes during murine lymphocytic choriomeningitis virus infection (LCMV). In this work, bone marrow-derived dendritic cells (BMDC) were stimulated with LCMV and EVs. The EVs showed CD11c, CD80, CD86 and MHC class I and II molecules (highly abundant) on their surfaces. However, vaccination with BMDC-derived EVs did not contribute to CD8^+^ T-cell cross-priming in vitro and did not protect the mice in a challenge trial. Thus, although dendritic cell (DC)-derived EVs activated anti-tumor immunity, in the case of LCMV, they did not activate antiviral cytotoxic T lymphocytes [75]. Fortunately, not all virus diseases behave in the same way. One example is the use of EVs to deliver specific microRNA to *Sus scrofa* cells inhibiting PRRSV virus infection. In particular, microRNAs were designed to target sialoadhesin or CD163, two main receptors involved in the attachment of viral particles and internalization [13]. The selected sequences expressed by means of the adenoviral vectors in cells were observed to be secreted in exosomes. Finally, cells exposed to microRNAs by adenoviral vector transduction and those exposed to exosomes both suppressed receptor expression at the mRNA and protein levels. Moreover, the PRRSV viral titer was reduced using both methods (rAd or exosomes), demonstrating not only a long-lasting effect but also effectiveness against different viral strains [76].

Proteomic studies identified PRRSV viral proteins in extracellular vesicles enriched from sera of convalescent pigs [77]. Thus, PRRSV proteins were detected in serum samples from only viremic animals and from animals who had previously been infected and were free of viruses (non-viremic) but not in controls. Moreover, immune sera from pigs previously exposed to PRRSV specifically reacted against exosomes purified from non-viremic pig sera in a dose-dependent manner. Reactivity was not detected when naïve sera were used in the assay. Moreover, EVs from convalescent sera were recognized similarly to how they were in the MLV vaccine Porcilis PRRSV (MSD Animal Health) in ELISA tests, giving statistically significant results when compared to PRRSV naïve sera used with EVs or MLV [77]. In addition, the same EVs were enriched using a mid-scale process and were tested in the first targeted pig trial using EVs from a viral disease. EV preparations enriched in high volumes of sera contained viral proteins and when injected into naïve pigs (up to 2 mg), they did not cause any secondary effects or clinical signs associated with PRRSV, indicating that the EV preparations were safe and virus free. In addition, PBMCs derived from vaccinated pigs using a prime–boost strategy (combining EVs and viral peptides) elicited antibody immune responses, but, importantly, they elicited cell-mediated immune responses that could be measured by IFN-γ secreting cells, which was not observed when using a classical peptide plus adjuvant approach. These results strongly suggest that EVs are an important tool for antigen discovery and an interesting vaccine platform against PRRSV [26].

EVs are gaining attention in regard to other veterinary viral diseases. The Marek’s disease virus (MDV), the causative agent of Marek’s disease (MD), is a complex pathology of chickens in which the main clinical signs are associated with paralysis, immune suppression, and T-cell lymphomagenesis. Serum EVs from CVI988 (Rispens)-vaccinated and protected chickens contained a high abundance of anti-tumor microRNAs in comparison to those animals who were not vaccinated. More importantly, mRNAs mapping the whole MDV genome were identified in those EVs from vaccinated chickens, providing important data regarding immune suppression caused by MDV and vaccine responses as well as new biomarkers associated with protection and susceptible animals [78].

EVs have been also evaluated in the context of ASFV. In animals infected with the naturally attenuated strain OURT 88/3, when the viremia disappeared from circulation, the serum EVs still contained viral proteins, such as p30 and p72, as detected by a bead-based flow cytometry assay in the absence of complete viral particles at 24 days post infection. Moreover, only one viral protein (VF602_ASFM2 Protein B602L) was detected through proteomic analyses despite the detection observed by flow cytometry. Interestingly, depending on the strain used for infection (naturally attenuated OURT 88/3 or laboratory deletion mutant Benin ΔMGF), the porcine protein cargo was different in the enriched EV fractions, and Benin ΔMGF had more differentially expressed proteins than OURT 88/3. These results suggest a new field to be explored in ASFV in order to determine different vaccination approaches based on EV proteins and nucleic acids, opening an opportunity for the evaluation of the new antigens found in ASFV EVs to be tested as vaccine candidates [79].

In conclusion, despite the limitations associated with purification methods (improvements required and high cost), the quantification of particular EV populations and the further exploration of adverse effects, EVs represent a new strategy to be explored for antigen discovery and vaccination. The few tests that have been conducted in the context of viral animal diseases have produced promising results, showing their potential to be exploited in this field.

## 7. Regulatory

The regulatory roadmap for the use of EVs as a new vaccine approach against animal diseases is ill-defined but most likely will follow a similar path as that being developed for human diseases [30,80]. Moreover, the framework of the major international regulatory agencies, such as those of the European Union (EMA) and the United States (FDA), will have to be followed. EVs are considered novel biological medicinal products, and, thus, new rules explicitly regulating EVs have not yet been envisioned. As EVs can be obtained from cells in their natural state or genetically modified (this could modify the mechanism of action (MoA) of EVs), the International Society of Extracellular Vesicles proposed a classification that should be taken into account and includes unmodified cells and native EVs (biological medicine), genetically manipulated cells but EVs with no trans-gene products (biological medicine), EVs that contain trans-gene products from manipulated cells (gene therapy products), and native EVs as drug delivery systems (biological medicine) [30].

Based on the previously mentioned references, the MoA and the non-active components of EV formulations need an in-depth characterization in order to identify any possible effects and actions of the final product [80]. Afterwards, potency assays should be performed to evaluate EV preparations, concluding that at the time of application for product licensing, significant data regarding the MoA, molecular fingerprints, and other additional information supported by clinical efficacy and safety must be provided to allow final approval. Of note, several clinical trials on EV vaccines and therapeutics are presently approved by the FDA (available online: clinicaltrials.gov/ct2/home) (March 2021), thus indicating that regulatory issues for using EV-based vaccines will rapidly move forward, having a relevant impact on health. In order to develop these vaccines under regulatory guidelines and to achieve commercial availability in terms of vaccines for animal health (Figure 1), the gaps in our knowledge must first be filled.

## 8. Concluding Remarks

This review highlights different vaccination strategies against animal diseases of veterinary importance. These include classical approaches, such as attenuated or inactivated pathogens, as well as more recent approaches, such as DNA and viral-vectored vaccines; new adjuvant formulations; DNA and peptide vaccines; and nanoparticle vaccines, including VLPs and EVs. EVs represent a novel and promising strategy, as they are able to cause direct antigen presentation to immune cells, contain specific pathogen proteins that could trigger the desired immune responses, and act as an adjuvant when used as vaccines. However, key gaps in the knowledge of EV vaccination, such as the scale-up of production, the discovery of protective antigens, the mode of action, the regulatory pathways for vaccine licensing, among others, need to be addressed before these vaccines can reach the market. The recent development of mRNA vaccines against SARS-CoV-2 is rapidly opening a new avenue for vaccines against infectious diseases, and an exploration of the combination of EV and RNA vaccination approaches is warranted.

## Figures and Tables

**Figure 1 viruses-13-01499-f001:**
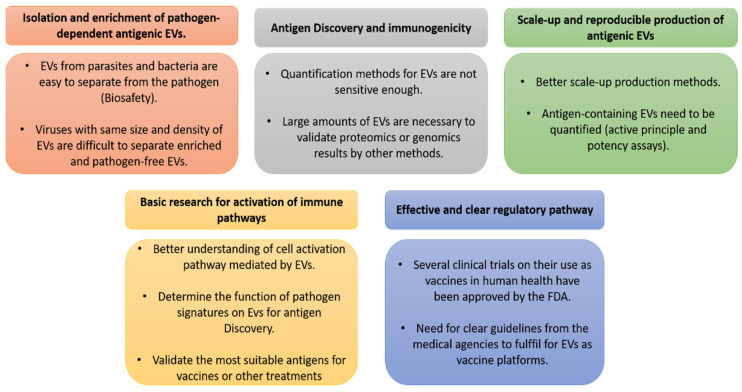
Key gaps in knowledge to be filled for the use of EVs as vaccines.

**Table 1 viruses-13-01499-t001:** Features, advantages and disadvantages of available vaccination approaches for animal viral diseases.

Vaccine Strategy	Starting Material	Require Adjuvant	Reversion to Virulence	Immunogenicity	Type of Immune Response	Advantages	Disadvantages	References
Live attenuated	Attenuated pathogen	Usually No	Possible	+++	Th1 and Th2	All antigens are in the original conformation	Could revert to virulent state	[7,8,9,15]
Loss of virulent factors and conserve immunogenicity	Genetic instability
Could be attenuated using mutagenic compounds	Loss of replication capacity, causing loss of immunogenicity
Wider presentation of antigens	Could spread to contacts due to replication
Different routes of administration including the natural ones	DIVA is difficult, as vaccinated and infected animals share similar antibodies
Affordable costs for veterinary pharmaceutical companies	
Inactivated and subunit vaccines	Peptides or proteins	Usually Yes	N/A	++	Mainly Th2 in most cases	Easy to produce at large scale (cost efficient)	Identification of protective epitopes takes time	[8,16,17,18]
Can be produced in different expression systems	Usually need adjuvant
Well-defined composition	Requires boosting
Primary immune responses (as well as cellular and antibody responses)	Usually weak immune responses.
Vectored vaccines	Genes encoding protective antigens	Usually No	N/A	++	Th1 and Th2	Production of the antigen in the cell of interest directly	Sometimes, there is pre-existing immunity against the viral vector	[8,9,17,19,20,21]
Very stable	Small genome size and need of helper virus during propagation in adenoviral vectors
Not adjuvanted	Poor immunity in some cases
Allows one to differentiate vaccinated from infected (DIVA)	Boosting required for full protection
Limited replicative capacity	Genetically considered modified organisms and potential health risk
Multiple epitopes can be included	
Nanovaccines (carriers)	Proteins or peptides and immunogenic carriers	Yes	N/A	+++	Possibly Th1 and Th2; further studies required	Protects the antigen and increases its immunogenic capacity	Identification of carriers to trigger specific immune responses	[22,23,24,25]
Allow antigen delivery using the mucosal route of immunization	For PGLA particles: low loading efficiency, hydrophobicity, fast burst release, high manufacturing cost, and scale-up difficulty
Scalable production of VLPs	Liposomes have low antigen loading and poor stability
Extracellular vesicles	Production of EVs from infected cells or producer cells	Not known	N/A	Need to be further studied	Possibly Th1 and Th2; further studies required	The cargo reflects the cell of origin	Production and scalability are difficult	[26,27,28,29,30]
Are able to self-present antigens (MHC molecules in their surface)	Antigen identification in samples needs further research
Can generate protective immune responses	Characterization of immune responses for each particular disease needs further research
Can be useful for different kind of diseases from cancer to infectious diseases	There is no clear regulatory frame (EMA/FDA) to move from research to industrial production and commercialization
Are able to pass the blood–brain barrier

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
