# Peer review of "Exosome-Based Vaccines: Pros and Cons in the World of Animal Health"

_viruses, 2021, doi:10.3390/v13081499_

Round 1
Reviewer 1 Report
Vaccination design is a complicated, evolving topic, maturing alongside our growing knowledge of how the immune system works, and influenced by recent technological advances and discoveries in the biotech space. Overall, this is a very nice review with an appropriate unique perspective on the role of novel EV-based vaccines for animals, especially because it includes a discussion of the pros and cons of various vaccination approaches, and because it highlights the grey-area within the regulatory space, and the novel regulatory implications that arise as a result of the unique biology of EV-based vaccines.
In fact, the manuscript would benefit significantly from clarifying these two important points as follows.
1) By providing a summary table with the pros and cons of each vaccination design approach (focusing on viruses), including key references. This detailed comparison of live attenuated vaccines, subunit vaccines, vector DNA vaccines, virus-like particles, and extracellular vesicles would provide a nice summary and highlight the key differences.
2) By expanding and elaborating upon the discussion of the regulatory framework. Human regulations are referenced but not described in any deep way. This should be elaborated upon in more detail. The content within Box 1 "Key gaps in knowledge to be fill for the use of EVs as vaccines" should contain far fewer words, highlight main talking points only within the figure, and the content of the figure should be moved into the main body of the manuscript, with more details given within the text. Much more detail can be given and references provided to back up the important statements about the difficulty to separate viruses and EVs, for example, as well as how to quantify EVs and determine the concentration.
Importantly, the manuscript requires a significant amount editing and proof-reading for English language problems, including rampant misspellings and poor grammar.
Reviewer 2 Report
This review article covers current available technology for vaccine development, and particularly focuses on extracellular vesicles for antigen delivery. It also covers the regulatory for EV use in the future.
I would recommend to include the following issues for EV-mediated vaccination. 1) because EVs are heterogeneous ; 2) harvesting EVs from culture is expensive, 3) the potential side effects and toxic effects of EVs for vaccination, such as induction of autoimmunity and others.
Round 2
Reviewer 1 Report
Overall, this is a nice revision and the authors have addressed the main concerns. The summary table and Box 1 are both much improved. The choice and quality of citations is very high in this review article and will be appreciated by the readership. There are still grammatical and spelling errors which must be corrected prior to publication. The formatting will also need to be improved.
Author Response
Dear Reviewer 1.
We thank your suggestions for English improvement and layout formatting. To make sure the article fit under the parameters of MDPI viruses we submitted the article to MDPI author services for those matters.
We believe this process improved the article to fulfil the requirements you were asking for MDPI Viruses publication.
Our best regards